# Rhesus-minus phenotype as a predictor of sexual desire and behavior, wellbeing, mental health, and fecundity

**Jaroslav Flegr**[1,2]*, **Radim Kuba**[2,3], **Robin Kopecký**[2]

**1** Department of Applied Neurosciences and Brain Imagination, National Institute of Mental Health, Klecany, Czech Republic, **2** Department of Philosophy and History of Science, Faculty of Science, Charles University, Prague, Czech Republic, **3** Department of Teaching and Didactics of Biology, Faculty of Science, Charles University, Prague, Czech Republic

\* flegr@cesnet.cz

## Abstract

### Background

Since its discovery in the 1930s, the effects of Rh phenotype on human health and wellbeing, with the exception of the effects of Rh-negativity of a mother on the risk of hemolytic anemia of Rh-positive children, has only rarely been studied. In the last few years, however, several studies have shown that Rh-negative subjects have worse health and performance in certain tests than their Rh-positive peers. Nothing is known about the effect of Rh phenotype on the quality of life of subjects as measured by a standard instrument.

### Methods

We hereby analyzed the data of 1768 male (24% Rh-negative) and 3759 female participants (23% Rh-negative) of an anonymous internet study using the partial Kendall test with the age and the population of the hometown of subjects controlled.

### Results

The results showed that the Rh-negative women, but not men, scored worse in wellbeing measured with the WHO-BREFF. The Rh-negative men scored worse in mental health-related variables and in their reported economic situation and the Rh-negative women scored better in physical health-related variables. Both the Rh-negative men and women reported higher sexual activity than their Rh-positive peers.

### Conclusions

The effects of the Rh phenotype were significant after the correction for multiple tests. However, they were usually weaker and less numerous than those of smoking, consuming alcohol, and high body mass index, which were used as a sort of internal control.

**Data Availability Statement:** All relevant data are available at Figshare https://doi.org/10.6084/m9. figshare.8216276 [24].

**Funding:** JF: 18-13692S Czech Science Foundation The funders had no role in study design, data

collection and analysis, decision to publish, or preparation of the manuscript.

**Competing interests:** The authors have declared that no competing interests exist.

# 1 Introduction

About 16% of the population in Europe (5% in Africa and 1% in Asia) are Rh-negative, i.e., they carry two copies of the allele of the RHD gene with large deletion, the d allele [1,2]. The gene codes a part of ammonium or $CO_2$ pump on the surface of erythrocytes; however, the biological function of this pump is unknown [3,4]. The protein coded by the functional allele D possesses an extremely strongly immunogenic epitope. Therefore, the Rh-positive progeny of Rh-negative mothers often died due to hemolytic anemia of newborns before the introducing of prophylactic treatment with anti-D immunoglobulins during pregnancy [5,6]. It has been suggested that this selection pressure is (or was) counterbalanced by another selection, for example, by selection in favor of heterozygotes [7] or in favor of Rh-negative subjects during the Paleolithic period [8]. With the exception of this immunological effect and sporadic reports about the possible association of Rh negativity with certain personality traits [9], no effects of the Rh genotype on human phenotype have been described. Recently, however, several studies suggested the existence of specific effects of Rh phenotype and genotype on human psychomotor performance [10–14] and health [15–17]. Generally, Rh-negative subjects have worse health, score worse in some performance tests, and express higher sensitivity to various negative environmental factors, such as *Toxoplasma* infection and smoking. Rh-positive heterozygotes were found likeliest to have the best health and performance relative to all other subjects, suggesting that the Rh-polymorphism is sustained in the population by balancing selection in favor of heterozygotes [16]. Until now, no data has been published regarding the effect of Rh phenotype on quality of life, for example, on wellbeing, measured with a standard psychological instrument.

In the present cross-sectional study, we search for the possible effects of Rh negativity on wellbeing, physical and mental health, fecundity, and sexual desire and behavior of 5,527 subjects, 3,759 women and 1,768 men. These subjects participated in a large internet study primarily dealing with the effect of the *Toxoplasma* infection on physical and mental health. To enable assessment of the relative strength of the observed effects, we also computed the effects of smoking, alcohol consumption, and being overweight (body mass index) on the same set of health- and wellbeing-related variables.

# 2 Material & methods

## 2.1 Subjects

The internet questionnaire was distributed as a Qualtrics survey. Subjects were invited to participate in the study using a Facebook-based snowball method [18]. Potential volunteers, mostly members of the "Lab Bunnies" community, an 18,000-member group of Czech and Slovak nationals willing to take part in evolutionary psychology experiments, and their Facebook friends, were invited (using about 10 different posts on the Lab bunnies timeline) to participate in an anonymous study about "magical thinking, superstitions, prejudices, religion and the relation between various environmental factors, and health and wellbeing." The questionnaire was also promoted in various electronic and printed media and TV–always without mentioning RhD, rhesus factor, or blood groups. Responders were not paid for their participation in the study; however, after finishing the 80-minute questionnaire, they were provided information about the results of related studies and their own results of several tests that were part of the questionnaire. At the first screen of the survey, the participants were given the following information and were asked to provide their informed consent to participate in the study: "The study is anonymous and obtained data will be used exclusively for scientific purposes. Your cooperation in the project is voluntary, and you can terminate it at any time by

closing this web page. You can also skip any uncomfortable questions; however, complete data is most valuable. If you agree to participate in the research press the 'Next' button". Only the subjects who provided their informed consent by pressing the button could participate in the study. Some pages of the questionnaire contained the Facebook share and like buttons. These buttons were pressed by more than 1,600 participants, which resulted in obtaining data from 12,600 responders in total between 27[th] May 2016 and 29[th] June 2018. The project, including the method of obtaining electronic informed consent to participate in this anonymous study from all participants, was approved by the IRB of the Faculty of Science, Charles University ("Komise pro práci s lidmi a lidským materiálem Přírodovědecké Fakulty Univerzity Karlovy")—No. 2015/07.

## 2.2 Questionnaires

The electronic survey consisted of several parts that concerned various unrelated projects on evolutionary psychology and psychiatry. In the present study, we inspected and analyzed only responses to the questions concerning health, wellbeing, number of children (a proxy of biological fitness), sexuality, and Rh phenotype. The responders were asked about their biological *sex*, *age*, body height and weight, and the size of the communities where they currently live (ordinal variable *urbanization*– 0: less than 1000 inhabitants, 1: 1–5 thousand inhabitants, 2: 5–50 thousand inhabitants, 3: 50–100 thousand inhabitants, 4: 100–500 thousand inhabitants, 5: more than 500 thousand inhabitants). They were asked about their ABO blood group (not used in the present study) and in the next question about their Rh factor. The possible responses were "I do not know, I am not sure", "negative (the rarer variant)", or "positive (the more frequent variant)"; the answer "I do not know, I am not sure" was *a priori* checked. As a benchmark for the relative importance of Rh phenotype on the health, wellbeing, and biological fitness, we looked for the associations of health, wellbeing, and fitness with three unrelated but well-known risk factors: *body mass index* (BMI) calculated from body height and body weight, frequency of *smoking*, i.e., how many cigarettes they smoke a day (ordinal scale 0: 0, 1: 00.1, 2: 0.1–1, 3: 1.1–3, 4: 3.1–10, 5: 11–20, 6: 21–40, 7: more than 40), and frequency of *alcohol consumption* "not to be allowed to drive a car for a while for this reason" (ordinal scale: 0-never, 1- maximally 1× a month, 2- maximally 2× a month, 3- maximally 4× a month, 4- maximally 2× a week, 5- every second day, 6- every day, 7- nearly all the time; in Czech, no measurable level of alcohol in blood is tolerated in drivers).

In another part of the questionnaire, the information concerning the following outcome health-related variables was collected from the responders: How they rate their physical health status in comparison with other people of the same age (*subjective physical health problems*: six points scale, anchored with 0: definitively better status, 5: definitively worse status), how they rate their mental health status in comparison with other people of the same age (*subjective mental health problems*: the same scale), how they rate their *family situation*, e.g., the quality of emotional support they can receive (0: poor, 5: excellent), and how they rate the *economic situation* of their family (0: poor, 5: excellent). To obtain more objective and concrete information on the health status of responders, they were also asked the following questions: how many kinds of *drugs prescribed* by a medical doctor were they currently taking, how many kinds of *drugs non-prescribed* by a doctor were they currently taking ("how many different herbs, food supplements, multivitamins, superfoods, etc. do you currently take per day"), how many times they visited their primary care doctor in the past 365 days ("not for prevention"), how many times they used *antibiotics* in the past 365 days, and how many different *medical specialists* they visited ("not for prevention") in the past 5 years. The *physical health problems score* was calculated as a mean of Z-scores of the last five variables. The responders were also requested

to rate how much they suffer from *anxieties*, *phobias*, *depression*, *mania*, *obsessions*, *auditory hallucinations*, *visual hallucinations*, and *headaches* using eight 0–100 scales. The *number of diagnosed* and the *number of undiagnosed mental health disorders* of responders, both checked on a list of 25 mental health disorders and epilepsy, was counted. The *mental health problems score* was calculated as a mean of Z-scores of the last 10 variables. Using a 0–100 scale the responders also answered the questions – how intensely they are *sexually attracted to men*, how intensely they are *sexually attracted to women*, (Z-score of the higher of these two responses was considered as the intensity of *sexual desire*). They were also asked with *how many men (women)* they already had sex ("vaginal, oral or anal"). To respond to these two questions, the participants used a 0–9 ordinal scale (0: 0, 1: 1, 2: 2, 3: 3, 4: 4, 5: 5–6, 6: 7–9, 7: 10–19, 8: 20 and more). The higher of these two responses (men vs. women) was considered to reflect the number of *preferred sex-sexual partners*. Similarly, the participants were asked with how many men (women) they already exchanged in *French kissing* (the same ordinal scale as previous questions)–the higher of these two responses (men vs. women) was considered to reflect the number of *preferred sex-French kissing-partners*. In the other part of the question-naire, the responders were asked to estimate how many minutes daily they spend engaged in various activities. The list of 22 activities also contained "*any form of sex*, including consuming pornography". A *score of sexual activity* was computed as the mean of Z-scores of the number of preferred sex sexual partners, preferred sex French kissing partners, and minutes spent engaged in any form of sex per day. The number of biological *children* was used as a proxy for biological fitness.

The Czech version of the standard 24-item WHOQOL-BREF (The World Health Organisa-tion Quality of Life Assessment Instrument–the abbreviated version of the WHOQOL-100) [19] was used for the assessment of the quality of life of responders. This instrument was trans-lated into Czech and standardized to the Czech population [20]. It monitors the general quality of life and its four specific domains: Physical health (the activity of daily living, dependence on medical substances and aids, energy and fatigue, mobility, pain and discomfort, sleep and rest, work capacity), Psychological (bodily image and appearance, negative feelings, positive feel-ings, self-esteem, spirituality/religion/personal beliefs, thinking, learning, memory, and con-centration), Social relationships (personal relationships, social support, sexual activity), and Environment (financial resources, freedom, physical safety and security, health and social care: accessibility and quality, home environment, the opportunity of acquiring new information and skills, participation in and opportunity for recreation/leisure activities, physical environ-ment – pollution/noise/traffic/climate, transport).

## 2.3 Statistical analyses

Before any analyses, records of all subjects who did not answer the question about Rh factor and of about 2% of subjects who provided a suspicious combination of answers to other ques-tions (too high/low body height, weight, age, an unrealistically high number of neuropsychiat-ric disorders, who answered all or nearly all questions by the same code, etc.) were filtered out.

The final set contained data from 6,602 subjects. However, not all of them responded to all questions concerning their health and quality of life. The distribution of all relevant (semi-con-tinuous and ordinal) variables was visually checked and then all secondary indices, see above, were computed. Distributions of some variables were asymmetric and the number of subjects in the RhD-negative subgroup was lower than in RhD-positive subgroup (app. 23%). For this reason, we used the partial Kendall test to search for the effect of RhD phenotype on all vari-ables. It is a nonparametric test insensitive to the character of data distribution, which can therefore be used to analyze imbalanced data sets. We used R v. 3.3.1 [21], package ppcor v. 1.1

[22] for computing the partial Kendall correlation. The correction for multiple tests was done using the Benjamini-Hochberg procedure with the false discovery rate pre-set to 0.20 [23].

All relevant data are available at Figshare https://doi.org/10.6084/m9.figshare.8216276 [24].

## 3 Results

The final data set contained responses from 425 (24.0%) Rh-negative men (mean age 35.9, SD 12.25), 1,343 Rh-positive men (mean age 36.2, SD 12.5), 864 (23.0%) Rh-negative women (mean age 36.7, SD 13.0), and 2,895 Rh-positive women (mean age 36.5, SD 12.8). No difference in age between men and women, or Rh-negative and Rh-positive subjects was significant ($p > 0.76$).

Many examined health and wellbeing-related variables had asymmetrical or multimodal distribution, and most of them correlated with age and the population of the hometown of the responders. Therefore, we analyzed the effects of Rh negativity, smoking, alcohol consumption, and high body mass index (BMI) with nonparametric partial Kendall correlation with age and the population of hometown as covariates. Table 1 shows that Rh-negative women reported worse quality of life as measured with WHOQOL-BREF, worse economic situations, and better physical health while the Rh-negative men reported worse economic situations, worse mental health and lower number of children. Both the Rh-negative men and women also reported higher sexual activity. Table 1, however, also shows that the effects of the other three variables, namely smoking, alcohol consumption, and BMI, were stronger and more

**Table 1. Relationship between Rh-negativity and other factors with wellbeing and various health-related variables.**

| | wellbeing (WHOQOL-BREF) | no. of children | family situation | economic situation | mental health problems score | physical health problems score | subject. mental health problems | subject. physical health problems | sexual activity | sexual desire |
|---|---|---|---|---|---|---|---|---|---|---|
| | | | | | **Men** | | | | | |
| Age | **0.080** | **0.529** | **0.082** | **0.077** | **-0.203** | **0.035** | **-0.063** | **-0.040** | **0.217** | **0.062** |
| Urbanization | -0.020 | **-0.152** | -0.053 | -0.018 | 0.069 | 0.034 | 0.033 | 0.024 | **0.068** | -0.013 |
| Rh-negativity | 0.001 | **-0.033** | -0.017 | **-0.045** | 0.036 | -0.003 | 0.033 | -0.004 | **0.036** | -0.024 |
| Smoking | **-0.052** | -0.006 | -0.011 | **-0.081** | 0.073 | 0.005 | 0.038 | 0.084 | 0.216 | -0.031 |
| Alcohol consumption | 0.016 | 0.000 | 0.031 | 0.043 | 0.023 | -0.063 | 0.006 | -0.061 | 0.072 | 0.026 |
| Overweight (BMI) | **-0.103** | **0.094** | -0.013 | 0.002 | **0.034** | **0.076** | **0.058** | **0.198** | **0.047** | **0.044** |
| | | | | | **Women** | | | | | |
| Age | -0.005 | **0.549** | -0.041 | 0.028 | **-0.142** | **0.085** | **-0.082** | **-0.088** | **0.108** | **-0.046** |
| Urbanization | -0.004 | **-0.196** | -0.033 | -0.015 | **0.068** | -0.007 | **0.028** | 0.014 | 0.012 | 0.014 |
| Rh-negativity | **-0.031** | -0.011 | -0.004 | **-0.024** | -0.002 | **-0.024** | -0.001 | -0.005 | **0.030** | 0.002 |
| Smoking | **-0.046** | **-0.055** | **-0.089** | **-0.114** | **0.080** | 0.003 | **0.028** | **0.041** | 0.238 | **0.018** |
| Alcohol consumption | 0.014 | **-0.090** | -0.022 | -0.021 | **0.062** | -0.018 | **0.030** | -0.046 | **0.129** | 0.014 |
| Overweight (BMI) | **-0.074** | **0.055** | -0.026 | **-0.054** | **0.032** | **0.094** | **0.036** | **0.247** | 0.012 | 0.014 |

The table shows Taus (for the factors age and urbanization) and partial Taus with age and urbanization controlled (for other factors). The sign and size of the Tau reflect the direction and strength of the associations between the factors listed in the headers of rows and the responses listed in the headers of columns. A negative partial Tau indicates a negative correlation between the factor and the response, e.g., worse wellbeing in older subjects, subjects living in larger settlements, Rh-negative subjects, smokers, alcohol-consuming subjects and subjects with a larger body mass index. Associations that remained significant after the correction for multiple tests are printed in bold.

**Table 2. Relationship between Rh-negativity and other factors with the variables used for computing wellbeing, mental health, physical health, and sexual activity scores.**

|  | Men | | | | Women | | | |
|---|---|---|---|---|---|---|---|---|
|  | Rh-neg. | smoking | alcohol | BMI | Rh-neg. | smoking | alcohol | BMI |
| Physical health wellbeing | 0.008 | **-0.051** | 0.018 | **-0.098** | -0.021 | -0.010 | **0.033** | **-0.083** |
| Psychological wellbeing | 0.002 | **-0.039** | -0.013 | **-0.076** | **-0.033** | **-0.043** | **-0.039** | **-0.055** |
| Social relationships wellbeing | 0.018 | -0.011 | 0.001 | **-0.054** | **-0.025** | **-0.045** | -0.002 | **-0.042** |
| Environment wellbeing | -0.019 | **-0.049** | **0.037** | **-0.084** | -0.012 | **-0.045** | **0.030** | **-0.030** |
| Drugs prescribed | -0.001 | 0.021 | **-0.066** | **0.135** | 0.001 | **-0.026** | **-0.056** | **0.132** |
| Drugs non-prescribed | 0.003 | -0.011 | -0.018 | 0.004 | **-0.046** | -0.020 | -0.002 | -0.021 |
| Practical doctor (365 days) | -0.029 | -0.007 | **-0.038** | **0.055** | -0.009 | 0.003 | -0.016 | **0.089** |
| Antibiotics (365 days) | -0.011 | -0.018 | **-0.037** | **0.066** | -0.008 | **0.029** | 0.007 | **0.055** |
| Medical specialists (five years) | 0.023 | -0.005 | **-0.047** | **0.044** | -0.012 | -0.008 | -0.002 | **0.062** |
| Anxiety | 0.026 | **0.055** | 0.025 | 0.013 | 0.014 | **0.060** | **0.073** | 0.002 |
| Phobia | 0.026 | 0.013 | **0.045** | 0.019 | 0.002 | **0.032** | **0.026** | **0.038** |
| Depression | 0.033 | **0.085** | **0.056** | **0.051** | 0.001 | **0.073** | **0.088** | **0.032** |
| Mania | 0.035 | **0.096** | **0.055** | -0.027 | 0.004 | **0.093** | **0.085** | 0.012 |
| Obsession | 0.013 | 0.031 | **0.049** | -0.020 | -0.024 | 0.022 | **0.056** | -0.010 |
| Auditory hallucinations | -0.004 | 0.030 | 0.028 | -0.010 | 0.002 | **0.041** | 0.025 | -0.012 |
| Visual hallucination | 0.015 | 0.023 | 0.026 | -0.002 | 0.016 | **0.033** | 0.011 | 0.011 |
| Headache | 0.024 | -0.015 | -0.004 | **0.058** | -0.022 | 0.020 | 0.015 | 0.016 |
| Diagnosed mental health dis. | **0.052** | **0.089** | -0.046 | **0.059** | -0.002 | **0.098** | -0.020 | **0.067** |
| Undiagnosed mental health dis. | 0.011 | **0.099** | **0.036** | **0.044** | -0.017 | **0.063** | **0.039** | **0.034** |
| Preferred sex sexual partners | 0.032 | **0.203** | **0.053** | **0.063** | **0.032** | **0.234** | **0.112** | 0.021 |
| Preferred sex French kissing partners | 0.032 | **0.224** | **0.093** | 0.025 | **0.042** | **0.247** | **0.145** | 0.004 |
| Time spent by sex | 0.024 | **0.057** | **0.055** | **0.062** | -0.020 | **0.052** | **0.079** | -0.009 |

The table shows partial Taus with age and urbanization controlled (see the legend for Table 1). Significant associations are printed in bold; no correction for multiple tests was done in this exploratory part of the study.

numerous. Table 2 shows the effect of Rh-negativity, smoking, alcohol consumption, and BMI on wellbeing subscales and the individual variables used for computing mental health problems score, physical health score, sexual activity score, and sexual desire.

## 4 Discussion

The present study showed that Rh negativity was associated with the impaired economic situations, and wellbeing and improved physical health of women and with the impaired economic situations and mental health of men. The strength of the observed associations was relatively low, especially in comparison with other negative factors such as smoking, consuming alcohol, and high BMI. Surprisingly, we also found a relatively strong positive association of Rh-negativity with the sexual activity of male and female responders. In Rh-negative men, the higher sexual activity (higher number of past sexual partners, past French-kissing partners, and larger part of a day usually spent by sexual activities, including consuming pornography) contrasted with the trend for their lower sexual desire (Tau = -0.024, p = 0.187); no such trend for sexual desire was observed in Rh-negative women (Tau = 0.002, p = 0.865). It can be only speculated whether the higher sexual activity of Rh-negative subjects is part of the transition from a slow to a faster life strategy in individuals as a result of worse health status and correspondingly lower life-expectancy [25,26].

### 4.1 Unanticipated results of the explorative part of the study

The Czech and Slovak populations contain about 16% of Rh-negative subjects. The frequency of such subjects, however, was much higher (about 23%) among the participants of the present study and even higher (25%) among participants of other internet studies, for example among the participants of our previous study about the sexual behavior of the Czech population [27]. It can be speculated that a fraction of subjects intentionally or unintentionally misreports their Rh phenotype. If this fraction is similar in Rh-negative and Rh-positive subjects (for example about 7%) it could result in a seemingly higher representation of the rarer phenotype, here the Rh-negative phenotype. Our records show that the prevalence of Rh-negative subjects in 2,611 participants of our experiments whose Rh phenotype have been examined serologically in our lab during the past 15 years is 22.2%. Therefore, the more probable explanation is that the personality of Rh-negative and Rh-positive subjects differ and the Rh-negative subjects are more interested in participating in our internet, as well as non-internet studies. It is worth mentioning that overrepresentation of RhD-negative subjects can be observed in practically all our studies, including the oldest one that investigates the effect of toxoplasmosis on psychomotor performance of blood donors at the blood transfusion department of the Institute of Hematology and Blood Transfusion, Prague. In that study, the prevalence of RhD-negative subjects was 27.2% (33.3% in men and 17.8% in women) [10].

The participants who reported consuming alcohol more frequently reported worse mental health; however, they also reported better physical health than other people. Either the alcohol consumption in moderate amount has a positive effect on physical health, or it is more likely that people with physical health problems consume alcohol less frequently or at least are reluctant to report that they consume alcohol even in an anonymous internet study.

All negative factors under study, possibly except BMI in women, had positive effects on sexual activity reported by participants. This phenomenon deserves future attention.

### 4.2 Limitations of the study

As discussed above (section 4.1), RhD-negative subjects were overrepresented among participants of the present study as well as in all past studies (both internet and non-internet) we had performed on self-recruited populations. The most likely explanation of this phenomenon is that the personality profile and therefore also willingness voluntarily participate in research differs between RhD-negative and RhD-positive subjects. Such differences could influence the results of all questionnaire studies: one can never be certain whether the observed effects of the RhD phenotype are due to a real difference in wellbeing and health or just a consequence of inclination to report a worse quality of life. To address this important issue, a personality questionnaire ought to be included in future internet questionnaires and the relation between wellbeing/health, personality, and RhD phenotype should be analyzed using a structural modelling technique, such as path analysis.

It must be emphasized, however, that the responders were blind to group assignment because the question about Rh factor was 'hidden' among 600 other questions (the time needed to complete the questionnaire was about 80 minutes) and the Rh-negative and the Rh-positive participants had no reason to respond to questions regarding their health and wellbeing differently unless they really felt differently about those issues.

It ought to be also noted that possible effects of personality on self-selection is a challenge not only for internet studies but for all studies, including those performed on so-called 'representative' samples. All studies performed in the past forty years are (or should be) based on self-selected populations, because participants must sign an informed consent form before starting their participation. In this consent form, they must be clearly informed that their

participation is voluntarily and they can terminate their involvement in it without giving any reason and at any time. It is highly probable that the personality profile of consenting and not-consenting population differs.

This study, together with nearly all studies that started before the year 2018, was not pre-registered. The authors of non-pre-registered studies could be suspected of publishing only a subset of results, here, for example, the results for the variables showing the significant association with Rh phenotype. However, the authors has already published or pre-registered several studies [17,28–30] https://osf.io/y5z64/ in which the effects of other factors on the same or a very similar set of wellbeing- and health-related variables were analyzed.

The effect of Rh negativity on the output variables was analyzed by comparing the value of output variables in Rh-negative subjects and Rh-positive subjects. However, Rh-positive subjects represent a mixture of Rh-positive homozygotes and heterozygotes. It was shown that the health of Rh-positive heterozygotes, but often not Rh-positive homozygotes, is better than that of Rh-negative homozygotes [16]. It would be important to repeat the study on an Rh-genotyped population in the future in order to avoid underestimating the strength of the effect of Rh negativity.

Participants of the study could represent a nontypical subpopulation because they knew their Rh phenotype and were willing to participate even in absence of any material compensation in a relatively time-consuming anonymous study. They could be more inquisitive or perhaps more altruistic than typical members of the general population. It must be emphasized, however, that both the cases (Rh-negatives) and the controls (Rh-positives) were recruited from the same population of inquisitive and altruistic subjects. Any possible untypicality of the sample therefore could not affect the main result of the study. Still, the study should be repeated with other populations in the future and one ought to be careful regarding the generalization potential of results of the present study.

The present study analyzed the information provided by the participants of the study. Both the information about their Rh phenotype and about their health status may be invalid for some, possibly many, people. As was confirmed by Monte-Carlo modeling [31], a stochastic noise that is inevitably present in the internet studies data would be able to cause only the false-negative result of studies (non-detection of existing associations) but not the false-positive results of a study (detection of non-existent associations). However, specific subpopulations of people, e.g., intentionally inaccurate responders, may have characteristic responses to the questions of their Rh phenotype as well as their wellbeing and health. Existence of such populations can also result in false-positive results of a study. It will be therefore necessary to repeat the study on the population of people with empirically examined Rh phenotype, or preferably genotype. Such studies have been already done for other traits demonstrating, for example, the association between Rh phenotype and performance in weight-holding and handgrip tests [14], intelligence [13], psychomotor performance [10–12], and several personality traits [32]. It must be mentioned, however, that a large binational cohort study found no difference in mortality of Rh-positive and Rh-negative subjects [33].

The prevalence of RhD-negative subjects differs between European, Asian, and African populations and even between populations of various European countries. One could thus speculate that ethnic difference between RhD-negative and RhD-positive participants of the study might be responsible for the observed differences. On the other hand, the ethnic composition of Czech population is extremely homogenous and the questionnaire was written in Czech, a difficult Slavic language relatively easily learned only by Slovaks, since Slovak is a closely related language. Among about 15,000 of members of the Lab Bunnies community (a group of volunteers willing to participate in questionnaire studies) there are practically no

foreign surnames. It is highly probable therefore that more over 99% of participants of the present study are ethnic Czechs or Slovaks.

### 4.3 Strengths of the study

A relatively large number of participants.

Absence of *a priori* knowledge of the participants about the subject of the study (the effect of Rh phenotype).

Standard questionnaire WHOQOL-BREF used to assess the quality of life of the responders.

## 5 Conclusions

The present study performed on a self-selected internet population had shown that the Rh-negative subjects report worse quality of life than the Rh-positive subjects. It also has been confirmed that they had worse mental health (men), better physical health (women), and worse economic situations. Rh-negative subjects also reported higher sexual activity than the Rh-positive subjects. The effects of Rh-negativity were less numerous and weaker than the effects of smoking, alcohol consumption, and high BMI. In fact, the partial Tau 2.4–3.6 correspond to Cohen f 3.8–5.7, which fall within a range of small (but not negligible) effects according to the Cohen classification [34]. It must be remembered, however, that total number of Rh-negative subjects in the human population is large and that Rh-negative subjects stay affected by they Rh negativity permanently. Therefore, if the effects observed in our sample are present in the general population, their total impact on public health could be larger than that of some formally stronger effects, such as the impact of alcohol consumption.

## Acknowledgments

We thank Charles Lotterman for the final revisions of our text.

## Author Contributions

**Conceptualization:** Jaroslav Flegr.

**Data curation:** Radim Kuba, Robin Kopecký.

**Formal analysis:** Jaroslav Flegr.

**Funding acquisition:** Jaroslav Flegr.

**Investigation:** Jaroslav Flegr, Radim Kuba, Robin Kopecký.

**Writing – original draft:** Jaroslav Flegr, Radim Kuba, Robin Kopecký.

**Writing – review & editing:** Jaroslav Flegr, Radim Kuba, Robin Kopecký.

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
