## [Decision Letter · Decision Letter 0]

8 Jun 2020

PONE-D-20-09062

Rhesus-minus phenotype as a predictor of sexual desire and behavior, wellbeing, mental health, and fecundity

PLOS ONE

Dear Dr. Flegr,

Thank you for submitting your manuscript to PLOS ONE. After careful consideration, we feel that it has merit but does not fully meet PLOS ONE’s publication criteria as it currently stands. Therefore, we invite you to submit a revised version of the manuscript that addresses the points raised during the review process.

We look forward to receiving your revised manuscript.

Kind regards,

Calogero Caruso, MD

Academic Editor

PLOS ONE

Journal Requirements:

Reviewers' comments:

Reviewer's Responses to Questions

**Comments to the Author**

1. Is the manuscript technically sound, and do the data support the conclusions?

Reviewer #1: Partly

2. Has the statistical analysis been performed appropriately and rigorously? 

Reviewer #1: No

3. Have the authors made all data underlying the findings in their manuscript fully available?

Reviewer #1: Yes

4. Is the manuscript presented in an intelligible fashion and written in standard English?

Reviewer #1: Yes

5. Review Comments to the Author

Reviewer #1: Like many other studies on the influence of blood group antigens on general health outcomes, it is a cross-sectional study.

The main weakness of the study is the low and most probably biased proportion of individuals reporting their Rh phenotype. The study was an online study, and the analysis is based on the self-disclosed Rh phenotype of the participants. The authors state that there were 12600 responders, 2% were filtered out due to suspicious answering pattern. Non-suspicious responders who disclosed their Rh phenotype numbered 6602. From these numbers, I calculate that only about 47% of participants did disclose their Rh phenotype. Was this as random sample? I do not believe. In the Czech population, the prevalence of the Rh negative phenotype is 16%, in the study sample the frequency was 23%. The authors note this discrepancy and describe that the fraction of Rh negative is in a similar range in their other studies even if checked by serology. They conclude that Rh-negative subjects are more interested in participating in their internet as well as non-internet studies. This assumption is probably correct, because the Flegr laboratory has high regard in the Czech and Slowak population for “explaining the function of Rh”. Assuming that (i) the participation in an internet survey depends on the economic and psychological situation and (ii) the likelihood to participate is heavily confounded by the Rh phenotype, any effect observed may as well be due to the selection bias.

Surprisingly, the authors mention this observation but did not include it in “limitations of the study”. In my opinion, a highly biased selection process is a major limitation.

A second weakness not considered by the authors is the possible association of Rh phenotype and ethnicity. The Rh negative phenotype is much more frequent in Europe than in Asia and Afrika. Immigrants will most likely be overrepresented in the Rh positive group and might differ in psychoeconomic factors due to their cultural background and possible discrimination. However, to my knowledge the prevalence of immigrants in the Czech republic is low and I would have expected a worse economic situation of (Rh positive) immigrants, just the opposite of the reported effect.

Apart from this likely bias, the authors did their best in the analysis of the obtained set of data.

I disagree with the conclusions. The statement “The present study has shown, probably for the first time, that the Rh-negative subjects have, or at least report, worse quality of life than the Rh-positive subjects” cannot be derived from data established ina highly biased collective.

A minor comment:

Ref 9: The citation is not correct, the reference is Cattle RB, Hundleby JD, Blood groups and personality traits, Am J Hum Genet 1972; 24(4): 485-486. It is a two pages letter in which the second page carries the name of the second author and the references. In addition, I cannot find a statement on the association of Rh negativity with personality traits as suggested by the authors' citation (it just states that these possible confounders were excluded in an older study)

6. PLOS authors have the option to publish the peer review history of their article (what does this mean?). If published, this will include your full peer review and any attached files.

Reviewer #1: No

---

## [Author Response · Author response to Decision Letter 0]

17 Jun 2020

Formated version of the folloving text has been also uploaded as .doc file

Reviewer #1: Like many other studies on the influence of blood group antigens on general health outcomes, it is a cross-sectional study.

The main weakness of the study is the low and most probably biased proportion of individuals reporting their Rh phenotype. 

The biased proportion of individuals (overrepresentation of Rh-negative subjects) could be a strength, not weakness, of this study. Ideal for a study such as ours would be a case-control design: a comparison between a group of Rh-negative subjects and a group of Rh-positive controls of the same size. We used a suboptimal design, which admittedly could miss some week effects of the Rh but – more importantly – could not identify nonexistent effects, for pragmatic reasons: the main subject of the research questionnaire were the effects of latent infection of Toxoplasma gondii infection, that is, not Rh. 

We included following paragraph into the Material and Methods section of the manuscript:

“Distributions of some variables were asymmetric and the number of subjects in the RhD-negative subgroup was lower than in RhD-positive subgroup (app. 23%). For this reason, we used the partial Kendall test to search for the effect of RhD phenotype on all variables. It is a nonparametric test insensitive to the character of data distribution, which can therefore be used to analyze imbalanced data sets.”

The study was an online study, and the analysis is based on the self-disclosed Rh phenotype of the participants. The authors state that there were 12600 responders, 2% were filtered out due to suspicious answering pattern. Non-suspicious responders who disclosed their Rh phenotype numbered 6602. From these numbers, I calculate that only about 47% of participants did disclose their Rh phenotype. Was this as random sample? I do not believe. 

We absolutely agree. We now include in the Limitations section of the manuscript the following: 

“Participants of the study could represent a nontypical subpopulation because they knew their Rh phenotype and were willing to participate even in absence of any material compensation in a relatively time-consuming anonymous study. They could be more inquisitive or perhaps more altruistic than typical members of the general population. It must be emphasized, however, that both the cases (Rh-negatives) and the controls (Rh-positives) were recruited from the same population of inquisitive and altruistic subjects. Any possible untypicality of the sample therefore could not affect the main result of the study. Still, the study should be repeated with other populations in the future and one ought to be careful regarding the generalization potential of results of the present study.” 

In the Czech population, the prevalence of the Rh-negative phenotype is 16%, in the study sample the frequency was 23%. The authors note this discrepancy and describe that the fraction of Rh negative is in a similar range in their other studies even if checked by serology. They conclude that Rh-negative subjects are more interested in participating in their internet as well as non-internet studies. This assumption is probably correct, because the Flegr laboratory has high regard in the Czech and Slowak population for “explaining the function of Rh”. 

Our laboratory is known for a study of the effects of Toxoplasma on human personality and behavior that has been going on for 25 years. We started studying the effects of Rh factors on human health only seven years ago and five years ago, we started to publish the first results of these studies in international journals (in English). Over ninety percent of our responders do not know that Rh is one of the many subjects our research team investigates. In fact, we already ran eight large internet studies and only one (not the present one) was promoted as a study of the effects of Rh factor on human personality, performance, and health. We have observed an overrepresentation of Rh-negative subjects in all the abovementioned studies, including a popular study on sexual behavior. About 60,000 subjects took this long questionnaire (that took over 100 minutes) and only a small fraction of them ever heard about Flegr and his research.

In the amended version of the manuscript, we included the following information: 

“The questionnaire was also promoted in various electronic and printed media and TV – always without mentioning RhD, rhesus factor, or blood groups.”

We also wrote:

“It is worth mentioning that overrepresentation of RhD-negative subjects can be observed in practically all our studies, including the oldest one that investigates the effect of toxoplasmosis on psychomotor performance of blood donors at the blood transfusion department of the Institute of Hematology and Blood Transfusion, Prague. In that study, the prevalence of RhD-negative subjects was 27.2% (33.3% in men and 17.8% in women) (Novotna et al. 2008).” 

Assuming that (i) the participation in an internet survey depends on the economic and psychological situation and (ii) the likelihood to participate is heavily confounded by the Rh phenotype, any effect observed may as well be due to the selection bias.

Surprisingly, the authors mention this observation but did not include it in “limitations of the study”. In my opinion, a highly biased selection process is a major limitation.

We have now included in the Limitations section the following paragraph:

“As discussed above (section 4.1), RhD-negative subjects were overrepresented among participants of the present study as well as in all past studies (both internet and non-internet) we had performed on self-recruited populations. The most likely explanation of this phenomenon is that the personality profile and therefore also willingness voluntarily participate in research differs between RhD-negative and RhD-positive subjects. Such differences could influence the results of all questionnaire studies: one can never be certain whether the observed effects of the RhD phenotype are due to a real difference in wellbeing and health or just a consequence of inclination to report a worse quality of life. To address this important issue, a personality questionnaire ought to be included in future internet questionnaires and the relation between wellbeing/health, personality, and RhD phenotype should be analyzed using a structural modelling technique, such as path analysis.” 

We also added:

“It ought to be noted that possible effects of personality on self-selection is a challenge not only for internet studies but for all studies, including those performed on so-called ‘representative’ samples. All studies performed in the past forty years are (or should be) based on self-selected populations, because participants must sign an informed consent form before starting their participation. In this consent form, they must be clearly informed that their participation is voluntarily and they can terminate their involvement in it without giving any reason and at any time. It is highly probable that the personality profile of consenting and not-consenting population differs.”

Moreover, we also added:

“It must be emphasized, however, that the responders were blind to group assignment because the question about Rh factor was ‘hidden’ among 600 other questions (the time needed to complete the questionnaire was about 80 minutes) and the Rh-negative and the Rh-positive participants had no reason to respond to questions regarding their health and wellbeing differently unless they really felt differently about those issues.”

A second weakness not considered by the authors is the possible association of Rh phenotype and ethnicity. The Rh negative phenotype is much more frequent in Europe than in Asia and Afrika. Immigrants will most likely be overrepresented in the Rh positive group and might differ in psychoeconomic factors due to their cultural background and possible discrimination. However, to my knowledge the prevalence of immigrants in the Czech republic is low and I would have expected a worse economic situation of (Rh positive) immigrants, just the opposite of the reported effect.

The referee is right. The ethnic composition of Czech population is rather homogenous due to the fact that in the past century and more, people tended to emigrated from Czechia rather than immigrate here. Moreover, the questionnaire was written in Czech, which is a difficult Slavic language spoken by few people who do not reside here for a long time. It is, however, closely related to Slovak. In future, Czech citizens of Vietnamese origin (and there is a sizeable community here) could bring some additional factors to control for similar studies. Unfortunately, Vietnamese surnames are currently almost absent in the database of our registered volunteers. We are therefore certain that over 99% of participants of the present study are either Czech or Slovak. 

We included the following paragraph into the Discussion section: 

“The prevalence of RhD-negative subjects differs between European, Asian, and African populations and even between populations of various European countries. One could thus speculate that ethnic difference between RhD-negative and RhD-positive participants of the study might be responsible for the observed differences. On the other hand, the ethnic composition of Czech population is extremely homogenous and the questionnaire was written in Czech, a difficult Slavic language relatively easily learned only by Slovaks, since Slovak is a closely related language. Among about 15,000 of members of the Lab Bunnies community (a group of volunteers willing to participate in questionnaire studies) there are practically no foreign surnames. It is highly probable therefore that more over 99% of participants of the present study are ethnic Czechs or Slovaks.”

Apart from this likely bias, the authors did their best in the analysis of the obtained set of data.

Thank you.

I disagree with the conclusions. The statement “The present study has shown, probably for the first time, that the Rh-negative subjects have, or at least report, worse quality of life than the Rh-positive subjects” cannot be derived from data established in a highly biased collective.

We toned down the conclusion. Now we write:

“The present study performed on a self-selected internet population had shown that the Rh-negative subjects report worse quality of life than the Rh-positive subjects.”

We also substituted: 

“Therefore, the total impact of these effects on public health could be larger than formally stronger effects of, e.g., alcohol.”

with

“Therefore, if the effects observed in our sample are present in the general population, their total impact on public health could be larger than that of some formally stronger effects, such as the impact of alcohol consumption.”

A minor comment:

Ref 9: The citation is not correct, the reference is Cattle RB, Hundleby JD, Blood groups and personality traits, Am J Hum Genet 1972; 24(4): 485-486. It is a two pages letter in which the second page carries the name of the second author and the references. In addition, I cannot find a statement on the association of Rh negativity with personality traits as suggested by the authors' citation (it just states that these possible confounders were excluded in an older study)

Thank you for finding this error in our bibliographic database. Now we included the correct citation: 

Cattell, R.B., Young, H.B., Hundleby, J.D. Blood groups and personality traits. Am J Hum Genet 16(4), 1964: 397-402. 

In their paper, the authors described a method of Rh assay and how they included Rh into their statistical models. In their results, they unfortunately only state that “Associations with Rh type are not included in the present report.”

---

## [Decision Letter · Decision Letter 1]

30 Jun 2020

Rhesus-minus phenotype as a predictor of sexual desire and behavior, wellbeing, mental health, and fecundity

PONE-D-20-09062R1

Dear Dr. Flegr,

We’re pleased to inform you that your manuscript has been judged scientifically suitable for publication and will be formally accepted for publication once it meets all outstanding technical requirements.

Kind regards,

Calogero Caruso, MD

Academic Editor

PLOS ONE

Additional Editor Comments (optional):

Reviewers' comments:

Reviewer's Responses to Questions

**Comments to the Author**

1. If the authors have adequately addressed your comments raised in a previous round of review and you feel that this manuscript is now acceptable for publication, you may indicate that here to bypass the “Comments to the Author” section, enter your conflict of interest statement in the “Confidential to Editor” section, and submit your "Accept" recommendation.

Reviewer #1: All comments have been addressed

2. Is the manuscript technically sound, and do the data support the conclusions?

Reviewer #1: Yes

3. Has the statistical analysis been performed appropriately and rigorously? 

Reviewer #1: Yes

4. Have the authors made all data underlying the findings in their manuscript fully available?

Reviewer #1: Yes

5. Is the manuscript presented in an intelligible fashion and written in standard English?

Reviewer #1: Yes

6. Review Comments to the Author

Reviewer #1: I thank the authors very much for considering my arguments and toning down some conclusions. I am still puzzled with the RhD neg overhang among the study participants, however, the authors could convince me that it is very unlikely to be due to a special stimulus on RhD neg subjects to participate in an important "RhD function laboratory study". The argument with the overrepresantation of RhD neg individuals even before the group turned to Rh is convincing in this respect.

7. PLOS authors have the option to publish the peer review history of their article (what does this mean?). If published, this will include your full peer review and any attached files.

Reviewer #1: No

---

## [Editor Report · Acceptance letter]

6 Jul 2020

PONE-D-20-09062R1 

Rhesus-minus phenotype as a predictor of sexual desire and behavior, wellbeing, mental health, and fecundity 

Dear Dr. Flegr:

I'm pleased to inform you that your manuscript has been deemed suitable for publication in PLOS ONE. Congratulations! Your manuscript is now with our production department. 

Kind regards, 

on behalf of

Prof. Calogero Caruso 

Academic Editor

PLOS ONE